# Phase Optimization for Multipoint Haptic Feedback Based on Ultrasound Array

**DOI:** 10.3390/s22062394

**Published:** 2022-03-20

**Authors:** Zhili Long, Shuyuan Ye, Zhao Peng, Yuyang Yuan, Zhuohua Li

**Affiliations:** Harbin Institute of Technology Shenzhen, Shenzhen 518055, China; longzhili@hit.edu.cn (Z.L.); shuyuanye@foxmail.com (S.Y.); 19s153210@stu.hit.edu.cn (Z.P.); hiruruku@outlook.com (Y.Y.)

**Keywords:** haptic feedback, phase optimization, pseudo-inverse algorithm, ultrasound array

## Abstract

Ultrasound-based haptic feedback is a potential technology for human–computer interaction (HCI) with the advantages of a low cost, low power consumption and a controlled force. In this paper, phase optimization for multipoint haptic feedback based on an ultrasound array was investigated, and the corresponding experimental verification is provided. A mathematical model of acoustic pressure was established for the ultrasound array, and then a phase-optimization model for an ultrasound transducer was constructed. We propose a pseudo-inverse (PINV) algorithm to accurately determine the phase contribution of each transducer in the ultrasound array. By controlling the phase difference of the ultrasound array, the multipoint focusing forces were formed, leading to various shapes such as geometries and letters, which can be visualized. Because the unconstrained PINV solution results in unequal amplitudes for each transducer, a weighted amplitude iterative optimization was deployed to further optimize the phase solution, by which the uniform amplitude distributions of each transducer were obtained. For the purpose of experimental verification, a platform of ultrasound haptic feedback consisting of a Field Programmable Gate Array (FPGA), an electrical circuit and an ultrasound transducer array was prototyped. The haptic performances of a single point, multiple points and dynamic trajectory were verified by controlling the ultrasound force exerted on the liquid surface. The experimental results demonstrate that the proposed phase-optimization model and theoretical results are effective and feasible, and the acoustic pressure distribution is consistent with the simulation results.

## 1. Introduction

Human–Computer interaction (HCI), which evolved from a general level of mouse–keyboard communication to an advanced stage of speech–gesture control, is a fundamental technology in the Internet of things. Haptic feedback, as an essential technology of HCI, can significantly increase sensing and perception when integrated with audio–visual communication [1]. Moreover, haptic feedback has widespread applications, such as in emotional interaction, teleoperation and medical training [2,3,4,5].

Currently, researchers and engineers who specialize in haptic development mainly focus on wearable haptic devices, which can provide touchable feedback [6]. Bouzit et al. designed the Master II-ND, which uses custom pneumatic actuators arranged in a direct-drive configuration between the palm and the thumb, index middle and ring fingers [7]. The glove is connected to a haptic control interface that reads its sensors and engages its actuators. Inrak Choi et al. developed a Wolverine system, which can withstand over 100 N of force between each finger and the thumb with a power consumption as low as 0.24 mWh for each braking interaction [8]. Integrated sensors are employed for both feedback control and user input. Specifically, time-of-flight sensors are used to determine the position of each finger, and an inertial measurement unit (IMU) is manipulated for the overall orientational tracking. Qin et al. designed and calibrated a new six-DOF (degree-of-freedom) haptic device [9,10]. It consists of a double parallel linkage, a rhombus linkage, a rotational mechanical structure and a grasping interface, which is capable of multifinger interactions. These tactile haptic devices have the advantages of strong feedback and high control accuracy. However, most of them are complex and cumbersome in structure. Moreover, there are unexpected haptic feelings due to the tactile sensors.

In order to overcome the shortcomings of tactile haptic feedback, some noncontact haptic feedback technologies have been proposed. Suzuki et al. developed a force feedback technique based on air pressure [11], which could be the basis of a handy untethered human interface. Gupta et al. designed the AirWave system based on an air vortex ring [12]. They showed through objective measurements that AirWave can achieve a spatial resolution of less than 10 cm at a distance of 2.5 m. They further demonstrated through a user study that this can be used to generate direct haptic stimuli in different regions of the human body. Rajinder et al. designed the AIREAL system based on a similar theory [13]. Combined with interactive computer graphics, AIREAL enables users to feel virtual 3D objects, experience free air textures and receive haptic feedback on gestures performed in the free space. However, haptic feedback based on air pressure has some disadvantages, such as the fixed direction of air injection and unsustainable pressure. Weiss et al. proposed the FingerFlux system based on electromagnet arrays [14]. FingerFlux allowed users to feel the interface before touching and could also create both attracting and repelling forces. Studies showed that users could feel vibration patterns up to 35 mm above the table and that FingerFlux could significantly reduce drifting when operating on-screen buttons without looking. However, the operator needed to stick a permanent magnet on the finger, which limited the effective sensing range. Tong et al. designed a novel magnetic levitation haptic device based on electromagnetic theory [15]. The users could directly sense virtual tissues by moving a magnetic stylus in the magnetic field generated by the coil array of their device. Yoichi et al. presented a new method of rendering aerial haptic images that uses femtosecond laser light fields and ultrasound acoustic fields [16]. They used femtosecond lasers to create plasma in the air and scanned it at a high speed to achieve various three-dimensional effects [17]. However, the workspace of the system was limited to only 1 cm^3^.

With distinct advantages of strong penetration, a low cost, low power consumption and controllability, haptic feedback based on ultrasound radiation has attracted more attention and interest than ever before [18]. Shinoda et al. firstly proposed noncontact midair haptic feedback based on an ultrasound transducer array. Phase control was used to focus the energy to form a stable pressure at the focal point, and the linear movement of the haptic feedback point was implemented by controlling the position change of the focal point [19,20]. Based on this technology, they further developed many new HCI devices, combining both ultrasound haptic feedback and a visual system [21,22,23,24,25]. Gavrilov et al. proposed the concept of control points to solve the multipoint focusing problem [26]. It was shown that the arrays made it possible to form the regions of action through a focused ultrasound with various necessary shapes and a sidelobe (or other secondary peak) intensity level acceptable for practical purposes. Using these arrays, it was possible to move the set of foci off the array axis to a distance of at least ±5 mm. In the framework of Gavrilov’s research, Tom Carter et al. implemented multipoint focusing based on an ultrasound array [27]. They invented an Ultrahaptics system to demonstrate a spatial and multipoint haptic feedback interaction. Through psychophysical experiments, they showed that feedback points with different haptic properties could be identified at smaller separations. They also showed that users could distinguish between different vibration frequencies of noncontact points with training. Long B. et al. analyzed the characteristic roots of the phase matrix and used the weighted Tikhonov regularization method to optimize the phase in order to reduce the power loss caused by the introduction of frequency modulation for haptic sensation [28]. With these, they then implemented the focus point distribution in various shapes. Georgios et al. developed the Haptogram system with high-frequency switching, which was designed to provide a point cloud haptic display via acoustic radiation pressure [29]. A tiled 2-D array of ultrasound transducers was used to produce a focal point that was animated to produce arbitrary 2-D and 3-D haptic shapes. Harman, a supplier of automotive systems and accessories, is working with UK start-up Ultrahaptics to develop a tactile feedback system. When the virtual button is pressed or turned in air, the driver will feel the corresponding tactile feedback. This noiseless feedback can be used in lane departure warning systems, front-vehicle collision warning systems and blind-spot detection systems. Tactile feedback means that the device can provide users with a feeling of touch through vibration, force and motion. When the user presses the button, it will pop up on the virtual panel, just as a real button would pop up on a physical panel. Many Android phones offer similar tactile feedback.

Noncontact haptic feedback, with so many competitive advantages, is an inevitable trend for the future of HCI. Noncontact haptic feedback based on the air vortex ring and electromagnetics has the shortcomings of a limited effective range and inconvenient control. In contrast, ultrasound arrays are more flexible due to a controlled phase and have a much larger effective range. Currently, researchers majoring in ultrasound haptic feedback are concentrating on single-point focusing, and existing discussions for multipoint focusing are rare at best. In fact, the process of attaining the phase difference of each ultrasound transducer is fundamental to generating various focus points, especially when a there is a large number of ultrasound arrays. Moreover, generating multiple forces is impossible without a sufficiently accurate calculation model. In this study, we propose an effective optimization approach for the phase calculation in multipoint haptic feedback, which was verified by experiments. Meanwhile, the white noise influences the accuracy of the haptic feedback. Therefore, we had to use several filter algorithms to enhance the signal-to-noise ratio. In addition, the signal transmission delay between the FPGA and each ultrasonic transducer also influences the accuracy of the haptic feedback. Thus, we used compensation methods by adding or subtraction a constant value in FPGA.

The paper is organized as follows: Section 2 establishes the mathematical model of the acoustic radiation force for an ultrasound array and introduces the phase control for single-point focusing. The phase-optimization model for multipoint haptic feedback is proposed in Section 3. Section 4 describes the phase optimization was verified through single-point and multipoint haptic feedback experiments using our self-developed ultrasound platform. Section 5 draws the conclusion.

## 2. Acoustic Radiation Force Model

An ultrasound transducer can emit continual mechanical vibrations at high frequency when it is driven by a voltage. We set the ultrasound array as *N* rows and *N* columns, denoted by *N* × *N* in our design, as shown in Figure 1, where *N* denotes any integer number.

As for the *i*-th ultrasound transducer, its ultrasound radiation pressure *p_i_* under the polar coordinates *r* (the radial coordinate) and *θ* (the angular coordinate, often called the polar angle) is expressed as [30].
(1)$$p_{i}\left(r,\theta ,t\right)=j\frac{\rho_{0}u_{0}a^{2}\omega }{2r}\cdot \frac{2J_{1}\left(ka\sin\theta \right)}{ka\sin\theta }\cdot e^{j\left(\omega t-kr\right)}$$
where *a* is the transducer radius, *ρ*_0_ is the median (air) density, *u*_0_ is the acoustic amplitude constant, *ω* = 2*πf*, in which *f* = 40 kHz is the ultrasound frequency, $k=\omega /c_{0}$ is the wave numbers where *c*_0_ is the ultrasound velocity in the air, and *J*_1_$\left(*\right)$ represents the first-order Bessel function.

The total ultrasound pressure at the location of *r* and *θ* in the *N* × *N* array is a linear superposition of the pressure of each transducer, as below
(2)$$P(r,\theta ,t)=\sum_{i=1}^{N^{2}}p_{i}(r,\theta ,t)$$

The root mean square (RMS) value of the ultrasound pressure is obtained over a period *T*,
(3)$$P_{rms}=\sqrt{\frac{1}{T}\int_{0}^{T}P^{2}dt}$$

By controlling the emitting phase of each transducer at different locations, the ultrasound pressure of each transducer can be focused on a certain point. The time of flight (TOF) of the ultrasound wave from the *i*-th transducer to the focal point is obtained as
(4)$$\Delta t_{i}=\frac{\sqrt{{\left(x-x_{i}\right)}^{2}+{\left(y-y_{i}\right)}^{2}+z^{2}}}{c_{0}}$$
where (*x*, *y*, *z*) is the location of the focal point, and *(x_i_*, *y_i_*, 0) is the location of the *i*-th transducer.

The phase difference *D_i_*, which signifies the delay of each transducer, is calculated as
(5)$$D_{i}=T-\mathrm{mod}\left(\Delta t_{i},T\right)$$
where mod (*) is the mode-taking operation.

Thus, the ultrasound pressure of each transducer is determined by substituting the phase difference *D_i_* into the radiation pressure pi according to (1), and then the focal pressure is obtained by the superposition of each transducer.

The simulation using MATLAB as shown in Figure 2 demonstrates two typical calculations with a single focal point, one in which the desired focal point was set to be 20 cm over the plane of the ultrasound array and the other in which they were at different locations, as shown in Figure 2a. Figure 2b shows the phase difference distribution of each transducer, as calculated by Expression (5). Figure 2c shows the normalized ultrasound pressure distribution, which means that the ultrasound pressure can be focused by controlling the phase difference of each transducer.

## 3. Optimization Model for Multipoint Focusing

### 3.1. Optimization Model Based Pseudo-Inverse (PINV)

In this section, we show that we can also control the phase difference of each transducer to generate multiple focusing points. In our study, the ultrasound transducer array was set to be 16 × 16. When many desired focus points exist, it is challenging to attain the phase difference of each transducer due to the high-dimensional matrix. For this, we propose the pseudo-inverse (PINV) approach to intelligently pinpoint the effective solutions, which is similar to an edge-detection algorithm.

With the phase difference *D_i_*, we can obtain the ultrasound pressure of the *i*-th transducer in the array, as
(6)$$p_{i}\left(r,\theta \right)=\left(j\frac{\rho_{0}\omega a^{2}}{2r}\cdot \frac{2J_{1}\left(ka\sin\theta \right)}{ka\sin\theta }\cdot e^{j\left(-kr\right)}\right)\cdot \left(u_{0}e^{-j\omega D_{i}}\right)$$
which can be defined as *p_i_* = *****H******_i_*·*u_i_*, where

$H_{i}=j\frac{\rho_{0}\omega a^{2}}{2r}\cdot \frac{2J_{1}\left(ka\sin\theta \right)}{ka\sin\theta }\cdot e^{j\left(-kr\right)}$ is constant if the location and radius of the *i*-th transducer are determined. Note that $u_{i}=u_{0}e^{-j\omega D_{i}}$ contains the amplitude and phase of the *i*-th transducer.

The total pressure of the transducer array *N* × *N* is obtained by the following linear superposition,
(7)$$P=\sum_{i=1}^{N^{2}}p_{i}=\sum_{i=1}^{N^{2}}H_{i}\cdot u_{i}$$

That is,
(8)$$P=\left[\begin{array}{cccc} H_{1} & H_{2} & \ldots & H_{N} \end{array}\right]\cdot \left[\begin{array}{c} u_{1}\\ u_{2}\\ \vdots \\ u_{N} \end{array}\right]$$

Assuming that there are *M* desired focal points being *P*_1_, *P*_2_, … and *P_M_*, we can obtain
(9)$$\left(\begin{array}{c} P_{1}\\ \vdots \\ P_{M} \end{array}\right)=\left(\begin{array}{ccc} H_{11} & \ldots & H_{1N}\\ \vdots & \ddots & \vdots \\ H_{M1} & \cdots & H_{MN} \end{array}\right)\times \left(\begin{array}{c} u_{1}\\ \vdots \\ u_{N} \end{array}\right),$$
which can be simplified as
*P**_M_* × 1 = *H**_M_* × *N* · u*N* × 1(10)
where *H_M_* × *N* represents the forward calculation of the ultrasound pressure of those *M* focus points and *N* transducers, u*N* × 1 is a complex matrix containing the amplitudes and phases of those *N* transducers, and P*_M_* × 1 is a matrix containing the ultrasound pressure amplitudes of those *M* focus points. Then, the optimization function of ultrasound pressure is determined as
(11)$$\underset{u}{\min}f\left(u\right)={\left\|Hu-P\right\|}^{2}$$

Our target was to determine the amplitude and phase u of each transducer so that the total ultrasound pressure ***Hu*** is closest to the desired pressure distribution ***P***.

Generally, the number of desired focal points is less than that of the transducers, i.e., *M* < *N_2_*. Since the matrix ***H*** is of full rank in row, its right inverse matrix exists. By formulating the right inverse of ***H***, the minimum norm solution of the Expression (11) is equal to
(12)$$\begin{array}{l} \min\;{\left\|u\right\|}_{2}\\ \mathrm{subject}\;\mathrm{to}\;Hu=P \end{array}$$

The solution is,
(13)$$u=H^{*T}{\left(HH^{*T}\right)}^{-1}P$$
where ***H***^**T*^ is the conjugate transpose of the matrix ***H***.

Thus, the phase difference u of each transducer can be determined by applying the pseudo-inverse of ***H*** when multiple desired focal points are set in the ultrasound array.

### 3.2. Calculations of Multipoint Focusing

By setting multiple focal points, the amplitude and phase of the transducers were obtained by calculating the optimization function (13).

The left and right columns in Figure 3 show the simulations when the numbers of the desired pressure points were 2 and 4, respectively. Figure 3a–d show the desired points, phase difference distribution, amplitude distribution and the RMS values of the ultrasound pressure, respectively. It can be seen that when the ultrasound pressures at the desired feedback points were uniform, they met the desired multipoint focusing forces.

### 3.3. Complex Focal Simulation

It can be deduced from Equation (13) that the solution as obtained by the PINV calculation was not affected by the position and the number of focal points. Therefore, in order to generate haptic feedback with more complex shapes, we set more focal points and then used the PINV approach to obtain the corresponding phases. The left, middle and right columns in Figure 4a show the desired focal geometries arranged in the forms of a circle, a rectangle and a triangle, respectively. Figure 4b–d present the corresponding phase difference distribution, amplitude distribution and ultrasound pressure distribution, respectively. It can be seen that the ultrasound pressure distribution can be constructed according to the desired geometries.

Similarly, the PINV algorithm was utilized to obtain the shapes of letters. The left, middle and right columns in Figure 5a demonstrate the desired focal letters of H, I and T, respectively, and the corresponding phase distribution, amplitude distribution and ultrasound pressure distribution are demonstrated in Figure 5b–d, respectively. Through these simulations, it was proved that the PINV can obtain a reasonable phase to form a desired pressure distribution in the shape of certain geometries and various letters.

### 3.4. Uniform Amplitude Optimization

In the objective optimization based on the PINV approach, since there was no constraint on the variable u, the controlled amplitudes for various transducers were different, as shown in Figure 3c, or Figure 4c or Figure 5c, although the phase differences were attained. In fact, it was difficult for the electrical control module to trigger multiple transducers under different voltage amplitudes. Therefore, it was necessary to further optimize the variable u. The optimization target was to find out the phase of each transducer with a uniform amplitude that would satisfy ***Hu*** = ***P***.

We propose the weighted iterative optimization algorithm as a method to achieve the uniform amplitude of each transducer. The weighting matrix ***W*** is expressed as
(14)$$u_{w}=WH^{*T}{\left(HWH^{*T}\right)}^{-1}P$$
where ***W*** is an *N* × *N* real positive definite matrix, which can be optimized by the following iterative procedure presented in Table 1.

The process of the uniform amplitude optimization for five-point focusing is demonstrated in Figure 6a, which shows the amplitudes without optimization. The amplitudes with 1, 3 and 7 iterations of the weighted optimization are shown in Figure 6b–d, respectively. It is shown that the amplitude of each transducer became increasingly uniform after several iterations.

The ultrasound pressure distribution without and with optimization are presented in Figure 7a,b, respectively. In Figure 7b, although some noises were amplified, as shown as the red dotted line, they were much smaller in magnitude than the pressure of the focal points. The ultrasound pressure amplitudes of the five focus points were nearly uniform, which is acceptable in the practical control.

Similarly, the weighted iterative algorithm was employed to optimize the amplitudes of circular, rectangular and triangular shapes. In Figure 8, the first row shows the ultrasound pressure distribution without optimization, and the second row shows the result with optimization. It is obvious that although some noises appear near the desired shapes, they are acceptable since the pressure profiles are in the same level.

## 4. Experiment

### 4.1. Experimental Platform

In order to verify the feasibility of the proposed phase-optimization approach, we conducted haptic feedback experiments including single-point, multipoint and dynamic trajectory verifications. Figure 9a shows the implementation procedure for the ultrasound haptic feedback from simulations in experiments. Firstly, the desired shape was determined and then imported into the PINV optimizer to obtain the initial phase. Then, the weighted iterative optimization presented in Section 3 was carried out to achieve the uniform amplitude. In the meantime, the phase was updated and then downloaded into the FPGA controller, which triggered the ultrasound array using the optimized phases.

Figure 9b shows the hardware architecture of the ultrasound control system. It consists of an ultrasound transducer array, an FPGA controller, a power amplification module, a computer and an acoustic measurement system. The ultrasonic array consists of 16×16 ultrasonic transducers arranged in the shape of a square. Each transducer has a high frequency of 40 kHz, and the emission angles are within ±30 degrees. The FPGA model XC6SLX9 was selected as the controller. The IX4427 chip was chosen to amplify the FPGA signal to 15 V. The transducer arrays were triggered in a certain phase sequence, as shown in Figure 9c. Finally, the ultrasound pressures were focused on the liquid surface in order to visualize the actual distribution of the ultrasound force. The distance between the ultrasound array and the liquid surface was 10 cm, and the dimensions of the ultrasound array were 2 × 2. The frequency modulation ranging from 40 to 300 Hz for each PWM signal was implemented to successfully generate the haptic stimulus sensing [30].

### 4.2. Single-Point Haptic Feedback

Figure 10a shows the liquid surface under the ultrasound pressure of a single point. We can observe that an obvious distortion existed on the liquid surface, meaning that the ultrasound force emitted by the transducer array was successfully focused with strong radiation pressure. Then, we put a palm over the transducer array to feel the haptic sensing, as shown in Figure 10b. When the palm moved downward, we could feel the force from haptic feedback on the center of the palm and felt the air column blowing toward the palm.

The location of the single-point haptic sensing can be dynamically modified if the controlled phases of the transducer array are in an appropriate combination. We switched the controlled phase of the ultrasound transducers continuously to achieve dynamic trajectory feedback, which generated upward, downward, leftward, rightward, circular, square and triangular movement trajectories. Figure 11 demonstrates that the ultrasound focal force was controlled as it dynamically moved in a square trajectory on the liquid surface.

### 4.3. Multipoint Haptic Feedback

Figure 12a,b show the two and three desired focal points, respectively. Figure 12c,d present the depression effects on the liquid surface for the experimental verifications corresponding to Figure 12a,b, respectively. We also found that there were two and three depressions on the liquid surface, proving that the ultrasound pressure produced by the transducer array was consistent with the desired distribution. Figure 13 demonstrates the ultrasound pressure distribution on the liquid surface with four geometric shapes, i.e., a line (shown as Appendix A), a circle, a rectangle and a triangle. It was verified that the depression distribution on the liquid surface matched the locations of the desired focal points well, indicating the validity of the phase solutions obtained by the optimization model.

### 4.4. Dynamic Trajectory Tactile Feedback Experiment

The continuous change diagram of the actual liquid level part of the rectangular track and the continuous change process of the concave point on the liquid surface with the switching of the focus point of the ultrasonic array are shown in Figure 14. In addition, the dynamic Line diagram is shown in the Appendix A.

In order to verify the effectiveness of tactile feedback, we finished with an experiment comprising 17 participants. The average age of participants was 23 years old. At first, we informed the participants about all the track types. Next, we showed the track types in random order and asked the participants to choose the track they felt. The final experimental results are shown in Table 2. From the results of Table 2, we can find that the results are in agreement with the simulations.

## 5. Conclusions

(1)To achieve the ultrasound-based noncontact haptic feedback, we investigated the relationship between the emission force and the controlled phase of each transducer in the ultrasound phase array. By the superposition of the ultrasound radiation force by appropriate phase combination, we successfully achieved the haptic feedback of complex shapes such as a circle, a square, a triangle and letters. The novel technology has great potential for application in haptic perception and HCI due to the advantages of a low price and low power consumption.(2)The mathematical model of the radiation pressure of the ultrasound array was deduced. The optimization function to intelligently search the optimal phase was established, and the PINV algorithm was introduced to effectively solve the control phase issue. To address the inconsistent amplitudes in the PINV solutions, a weighted iterative optimization approach was proposed to further enhance the amplitude of the ultrasound array, making the electrical driving module trigger the transducer array in a uniform voltage. With these, we carried out a simulation to visualize the multipoint haptic feedback of complex shapes such as a circle, a rectangle, a triangle and letters.(3)For the experiment, we built the control and driving systems based on an FPGA controller. The ultrasound focal force on the liquid surface was tested. Experimental verification of the single- and multipoint and the square dynamic trajectory was conducted to visualize the corresponding ultrasound pressures and focusing distributions. Experimental results prove that the proposed phase optimization and the electrical control system are feasible options for ultrasound-based haptic feedback.

In future research, we would like to investigate various aspects of ultrasound-based haptic feedback, including, but not limited to, quantification measurement of the acoustic pressure at different angles and distances, irregular amplitude control to array transducers and tactile stimulation to human body.

## Figures and Tables

**Figure 1 sensors-22-02394-f001:**
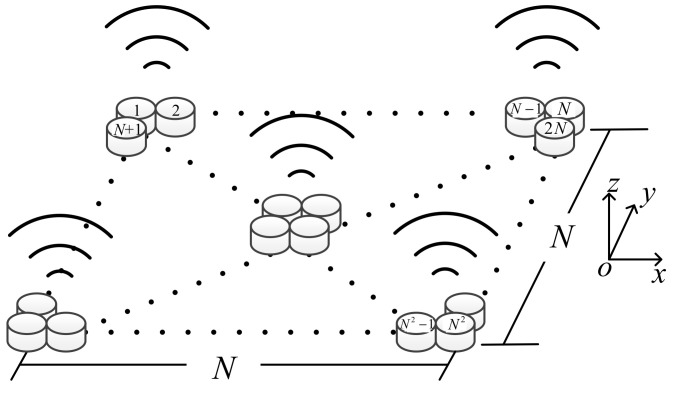
*N × N* ultrasound transducer array.

**Figure 2 sensors-22-02394-f002:**
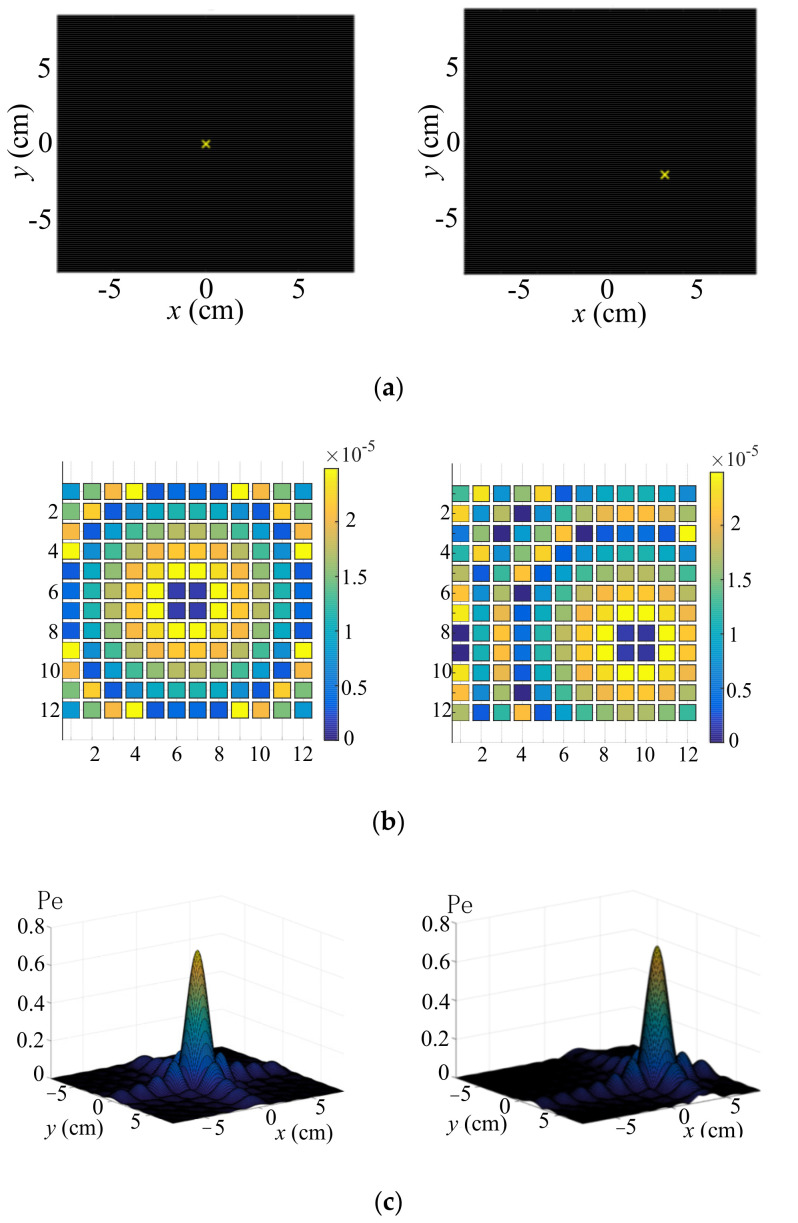
Simulation of single focal point. (**a**,**b**) are the phase difference distribution of each transducer. (**c**) is the normalized ultrasound pressure distribution.

**Figure 3 sensors-22-02394-f003:**
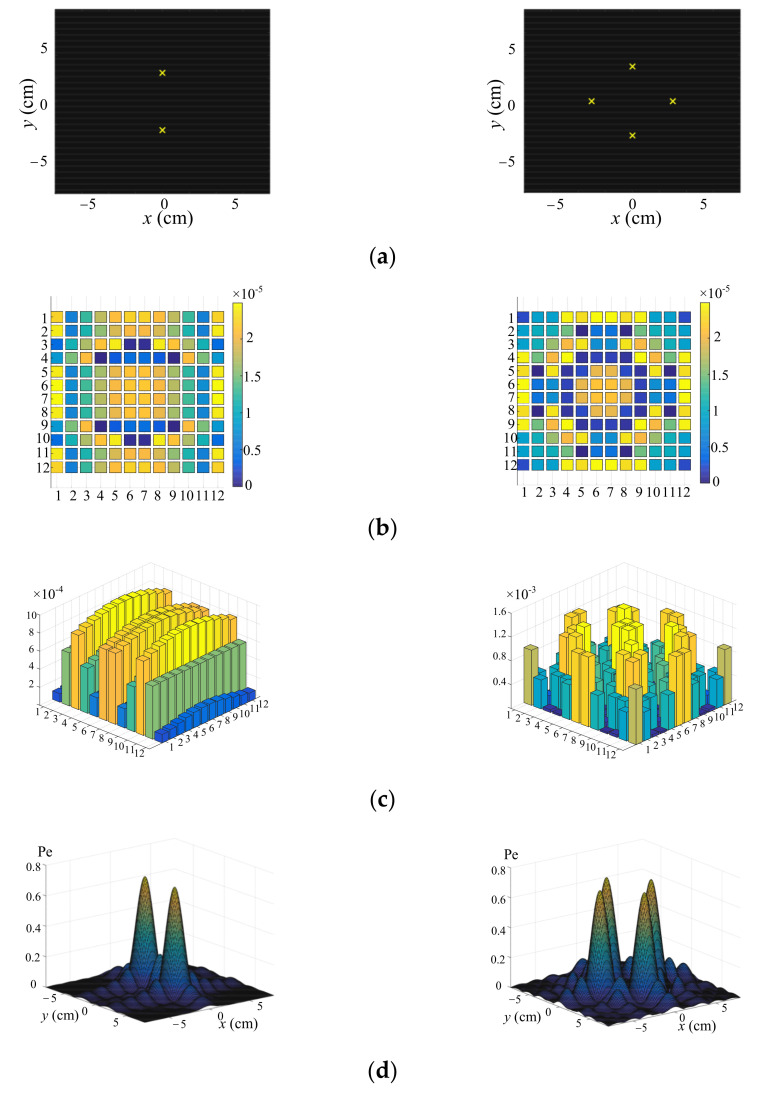
Simulation of multipoint focusing. (**a**–**d**) show the desired points, phase difference distribution, amplitude distribution and the RMS values of the ultrasound pressure, respectively.

**Figure 4 sensors-22-02394-f004:**
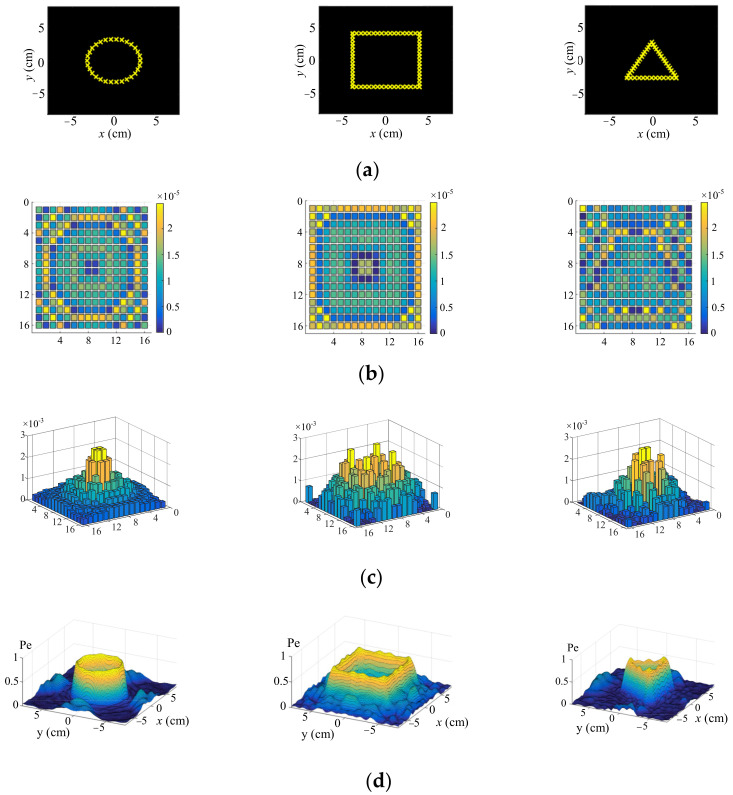
Simulation for complex geometries. (**a**) is the desired focal geometries arranged in the forms of circle, rectangle and triangle, respectively. (**b**–**d**) present the corresponding phase difference distribution, amplitude distribution and ultrasound pressure distribution, respectively.

**Figure 5 sensors-22-02394-f005:**
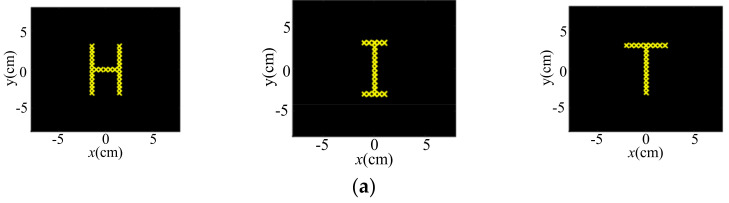
Simulation for letters H, I and T. (**a**) demonstrates the desired focal letters of H, I and T, respectively. (**b**–**d**) show the corresponding phase distribution, amplitude distribution and ultrasound pressure distribution, respectively.

**Figure 6 sensors-22-02394-f006:**
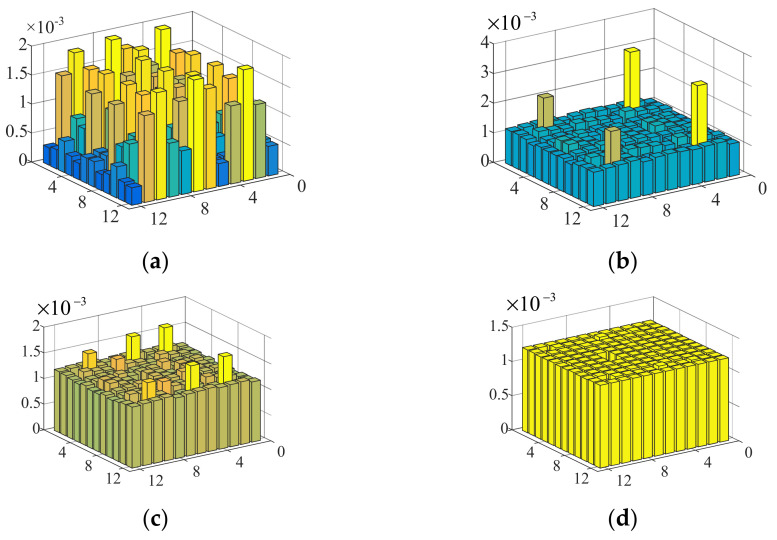
Iterative optimization for uniform amplitude. (**a**) Amplitude with 0 iterations (without optimization). (**b**) Amplitude with 1 iterations. (**c**) Amplitude with 3 iterations. (**d**) Amplitude with 7 iterations.

**Figure 7 sensors-22-02394-f007:**
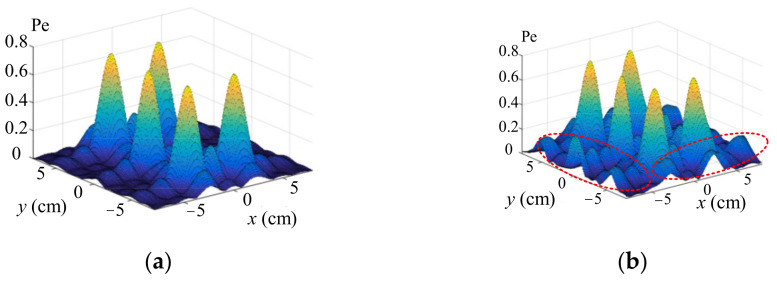
Ultrasound pressure distributions without and with optimization. (**a**) Without optimization. (**b**) With optimization.

**Figure 8 sensors-22-02394-f008:**
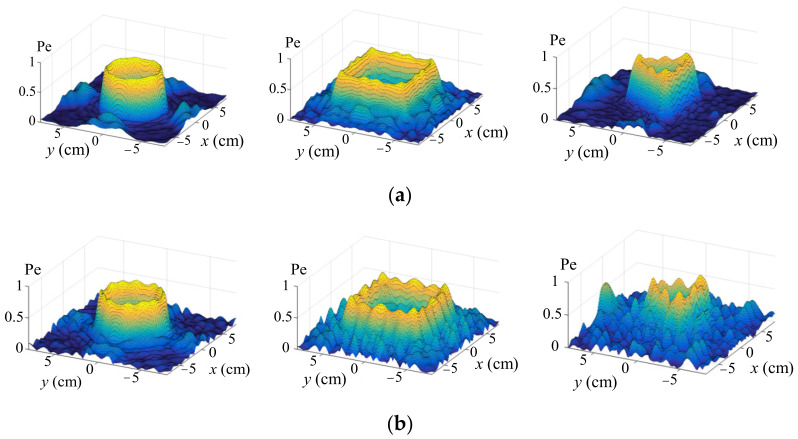
Pressure amplitude comparison with and without optimization. (**a**) is the ultrasound pressure distribution without optimization, and (**b**) is the result with optimization.

**Figure 9 sensors-22-02394-f009:**
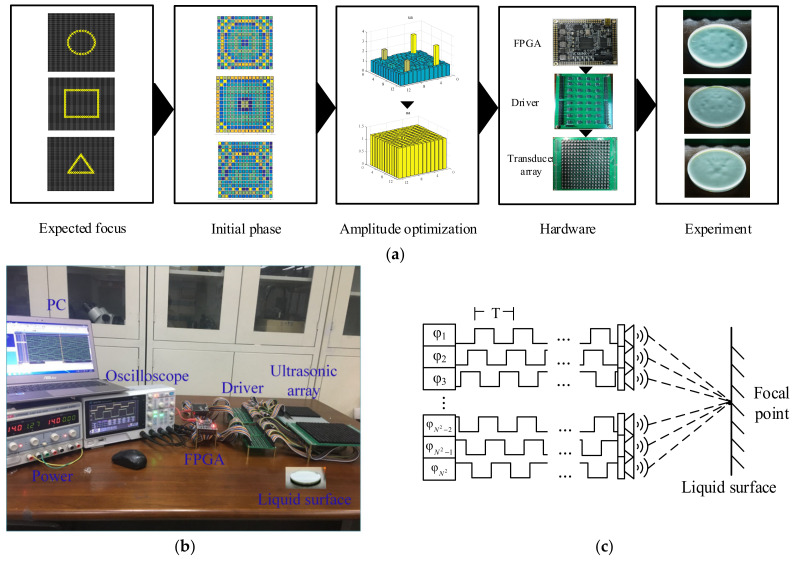
Experimental platform. (**a**) is the implementation procedure for the ultrasound haptic feedback from simulations in experiments. (**b**) shows the hardware architecture of the ultrasound control system. (**c**) shows that the transducer arrays are triggered in a certain phase sequence.

**Figure 10 sensors-22-02394-f010:**
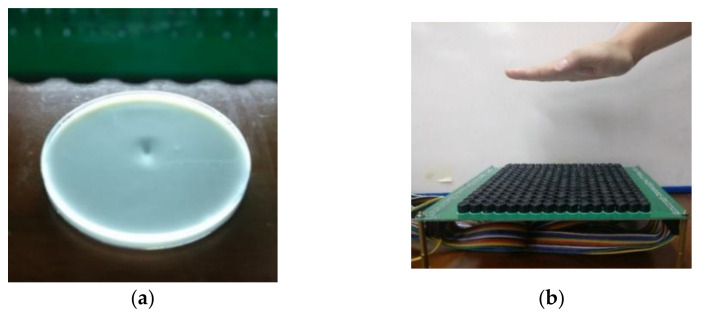
Single-point haptic feedback. (**a**) Liquid surface of single-point focus. (**b**) Haptic sensing to the palm.

**Figure 11 sensors-22-02394-f011:**
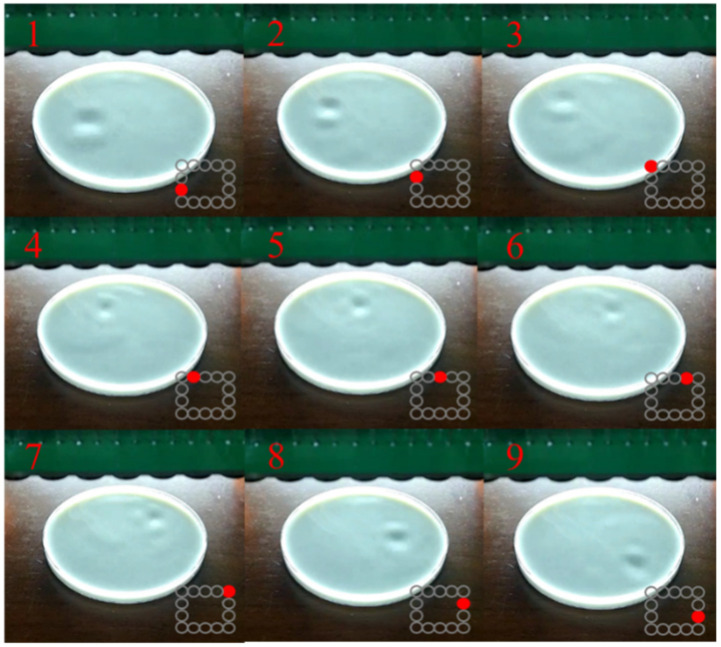
Dynamic movement of the ultrasound focal force.

**Figure 12 sensors-22-02394-f012:**
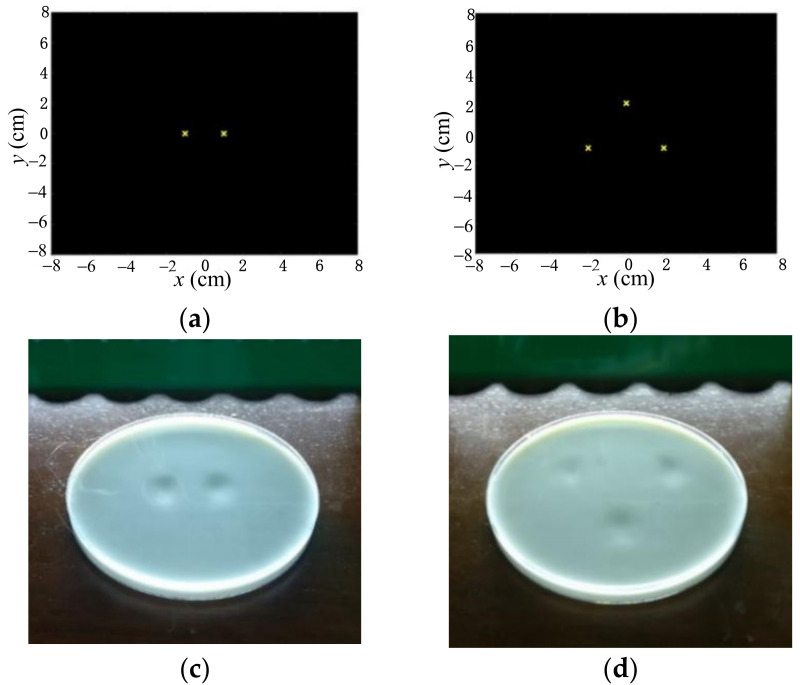
Ultrasound pressure with multiple points on liquid surface. (**a**) Desired two-point focuses. (**b**) Desired three-point focuses. (**c**) Two-point pressure distribution. (**d**) Three-point pressure distribution.

**Figure 13 sensors-22-02394-f013:**
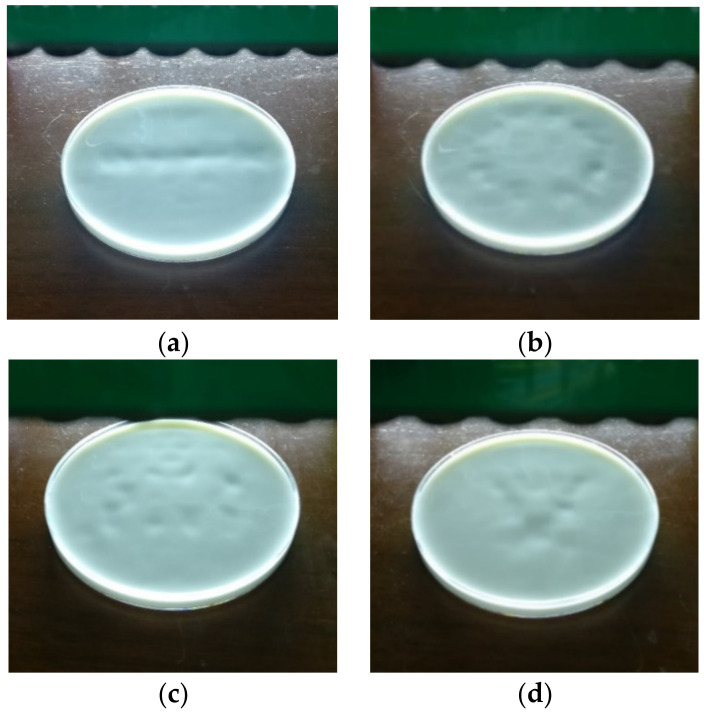
Ultrasound pressure with geometric shapes on liquid surface. (**a**) Line. (**b**) Circle. (**c**) Rectangle. (**d**) Triangle.

**Figure 14 sensors-22-02394-f014:**
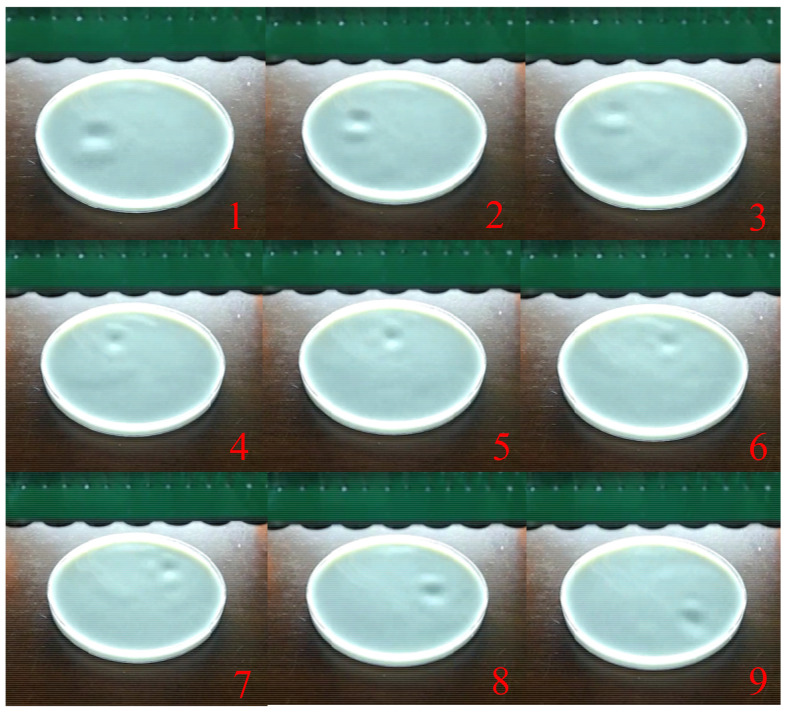
Dynamic rectangular partial continuous trajectory.

**Table 1 sensors-22-02394-t001:** Weighted iterative optimization algorithm.

Step 0: Initialize ***W*** *= **I***, ***I*** is the identity matrix.
Step 1: compute***u**_w_* and *η* by the following expression,
$u_{w}=WH^{*T}{\left(HWH^{*T}\right)}^{-1}P$
$\eta =\frac{\sum_{i=1}^{N^{2}}{\left|u_{i}\right|}^{2}}{N^{2}U_{\max}^{2}}\times 100\%$
If *η* is sufficient, then go to Step 3.
Otherwise, update ***H****^*T^* as,
***H***^**T*^ = ***WH***^**T*^
Step 2: Evaluate the updated the weighting matrix ***W***,
$W\left(m,n\right)=\left\{\begin{array}{cc} \frac{1}{\left|uwn\right|} & \mathrm{for}\;m=n;\\ 0 & \mathrm{otherwise}; \end{array}\right.$
Go to step 1.
where {*u_wn_*, *n* = 1, 2, …, *N*^2^} contains the elements of the vector *u_w._*
Step 3: The excitation vector ***u*** = ***u****_w_*.

**Table 2 sensors-22-02394-t002:** Effective identification of different dynamic trajectories.

		Forecast	Up Stroke	Down Stroke	Left Stroke	Right Stroke	Circular	Rectangle	Triangle	Correct Rate
	Actual									
Number										
Up stroke			17	0	0	0	0	0	0	100%
Down stroke			0	17	0	0	0	0	0	100%
Left stroke			0	0	17	0	0	0	0	100%
Right stroke			0	0	0	17	0	0	0	100%
Circular			0	0	0	0	13	3	1	76%
Rectangle			0	0	0	0	6	6	5	35%
Triangle			0	0	0	0	7	1	9	53%

## Data Availability

Not applicable.

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
