# Peer review of "Phase Optimization for Multipoint Haptic Feedback Based on Ultrasound Array"

_sensors, 2022, doi:10.3390/s22062394_

Round 1

Reviewer 1 Report

This paper presents an interesting work on phase optimization for multipoint haptic feedback based on ultrasound array. This work has some potential in terms of modelling and implementation. However, testing is lagging.

  1. In general, ultrasound haptic technology can be used to feel haptic feedback against the hands without any physical contact with a device and this could be an alternative for many applications since pandemic. The authors didn’t pay much attention on that. I highly recommend to highlight more application perspectives.
  2. Distortions, white signals, noises can be major issues in ultrasound haptic technology. I highly recommend to introduce in the introduction with pros and cons.
  3. In Acoustic radiation force model: can you say anything about latency? Perhaps you could consider TOF
  4. Figure 2(c) left is not visible. You can scale up
  5. This actuation in PINV similar to edge detection algorithm. How it is different from edge detection?
  6. A minor comment: I have a feeling that Figure 9 is out of the margin of the figure.
  7. To demonstrate hardware architecture showing in Figure 9, it would be better to upload a video to demonstrate the experiment. It could be via the Journal, or any web-based repository.
  8. The authors present that they put palm over the transducer array to feel the haptic force feedback. But it is not quantified. Nothing presents in results. How many participants? How did you collect data? Qualitative or quantitative? Did you get your ethical approval? Nothing is reported. It should be reported on the paper if you would like to say anything about how humans feel it.

Author Response

Thank you for your comments! 

Reviewer 2 Report

In this part I think you forgot a reference : "technologies are proposed. Suzuki et al. developed" . Suzuki et al should have a reference.

Just bellow Figure 1. Can you please add a short phrase to explain why you picked an array of N rows and N columns.

I know everybody wants to save space, but I would recommend to have Picture 2 enlarged so that you can properly display c)Ultrasound pressure distribution. Or is the Picture 2 correctly display ? 

Figure 7a and b look the very similar . Again , how should the reader compare 2 important images if he can't see them properly ? Maybe mark some differences. Provide a simple pixel by pixel algorithm to compare them and give us the differences in %. 

Can you please make some statements or find some citations on how safe (health-related) it is to use such interfaces in the future ? 

I would love to see some practical examples (even without the theoretical support) on how or where such interaction methods could be useful.  It is always very good to do things because we can, but I strongly encourage research with a tangible and concrete purpose.

Author Response

Thank you for your comments!

Reviewer 3 Report

The article "Phase Optimization for Multipoint Haptic Feedback Based on Ultrasound Array" presents an investigation of the phase optimization for multipoint haptic feedback based on ultrasound array and the corresponding experimental verification. In this reviewer’s opinion, the paper needs improvements:

1) If possible insert a comparative table of the proposed technique with other techniques.

2) Insert more statical details about the experimental verification.

3) What is the model of ultrasonic transducers? What are the dimensions of the ultrasonic transducers array?

Author Response

Thank you for your comments!

Round 2

Reviewer 1 Report

The authors managed to address my comments and I am happy with current version of the manuscript for possible publications.